# LiMAx Prior to Radioembolization for Hepatocellular Carcinoma as an Additional Tool for Patient Selection in Patients with Liver Cirrhosis

**DOI:** 10.3390/cancers14194584

**Published:** 2022-09-21

**Authors:** Catherine Leyh, Niklas Heucke, Clemens Schotten, Matthias Büchter, Lars P. Bechmann, Marc Wichert, Alexander Dechêne, Ken Herrmann, Dominik Heider, Svenja Sydor, Peter Lemmer, Johannes M. Ludwig, Josef Pospiech, Jens Theysohn, Robert Damm, Christine March, Maciej Powerski, Maciej Pech, Mustafa Özcürümez, Jochen Weigt, Verena Keitel, Christian M. Lange, Hartmut Schmidt, Ali Canbay, Jan Best, Guido Gerken, Paul P. Manka

**Affiliations:** 1Department of Gastroenterology, Hepatology and Transplant Medicine, University Hospital Essen, 45147 Essen, Germany; 2Department of Gastroenterology, Hepatology, and Infectious Diseases, Otto-von-Guericke University Magdeburg, 39120 Magdeburg, Germany; 3Department of Internal Medicine, University Hospital Knappschaftskrankenhaus Bochum, Ruhr University Bochum, 44892 Bochum, Germany; 4Central Laboratory, University Hospital Essen, University Duisburg-Essen, 45147 Essen, Germany; 5Department of Internal Medicine 6, Paracelsus Medical University Nürnberg, 90419 Nürnberg, Germany; 6Department of Nuclear Medicine, University Hospital Essen, 45147 Essen, Germany; 7Department of Mathematics and Computer Science, Philipps University of Marburg, 35032 Marburg, Germany; 8Institute for Diagnostic and Interventional Radiology and Neuroradiology, University Hospital Essen, 45147 Essen, Germany; 9Department of Radiology and Nuclear Medicine, Otto-von-Guericke University Magdeburg, 39120 Magdeburg, Germany; 10Department of Internal Medicine II, LMU University Hospital Munich, 81377 Munich, Germany

**Keywords:** hepatocellular carcinoma, HCC, LiMAx^®^, enzymatic liver function test, selective internal radiotherapy, radioembolization

## Abstract

**Simple Summary:**

Radioembolization is a well-established therapeutic option for patients with advanced hepatocellular carcinoma. However, patients with advanced tumor disease and presence of liver cirrhosis often present a borderline liver function and are at risk for post-interventional hepatic decompensation. The aim of our study was to evaluate the ability of the LiMAx^®^, a non-invasive test for liver function assessment, in predicting post-therapeutic hepatic deterioration and thus improve patient selection prior radioembolization.

**Abstract:**

Background and Aims: Radioembolization (RE) has recently demonstrated a non-inferior survival outcome compared to systemic therapy for advanced hepatocellular carcinoma (HCC). Therefore, current guidelines recommend RE for patients with advanced HCC and preserved liver function who are unsuitable for transarterial chemoembolization (TACE) or systemic therapy. However, despite the excellent safety profile of RE, post-therapeutic hepatic decompensation remains a serious complication that is difficult to predicted by standard laboratory liver function parameters or imaging modalities. LiMAx^®^ is a non-invasive test for liver function assessment, measuring the maximum metabolic capacity for 13C-Methacetin by the liver-specific enzyme CYP 450 1A2. Our study investigates the potential of LiMAx^®^ for predicting post-interventional decompensation of liver function. Patients and methods: In total, 50 patients with HCC with or without liver cirrhosis and not amenable to TACE or systemic treatments were included in the study. For patients prospectively enrolled in our study, LiMAx^®^ was carried out one day before RE (baseline) and 28 and 90 days after RE. Established liver function parameters were assessed at baseline, day 28, and day 90 after RE. The relationship between baseline LiMAx^®^ and pre-and post-interventional liver function parameters, as well as the ability of LiMAx^®^ to predict hepatic decompensation, were analyzed. Results: We observed a strong association between baseline LiMAx^®^ and bilirubin, albumin, ALBI grade, and MELD score. Patients presenting with Child–Pugh score B 28 days after RE or with a deterioration in Child–Pugh score by at least one point had a significantly lower baseline LiMAx^®^ compared to those with Child–Pugh score A or with stable Child–Pugh score. The ability of LiMAx^®^ to predict hepatic decompensation after RE was determined using ROC curve analysis and was compared to MELD score and ALBI grade. LiMAx^®^ achieved a substantial AUC of 0.8117, comparable to MELD score and ALBI grade. Conclusion: Patients with lower LiMAx^®^ values at baseline have a significantly increased risk for hepatic decompensation after RE, despite being categorized as Child–Pugh A. Therefore, LiMAx^®^ can be used as an additional tool to identify patients at high risk of post-interventional hepatic failure.

## 1. Introduction

Hepatocellular carcinoma (HCC) is among the leading causes of cancer-related mortality worldwide. During the last decades, increasing incidence rates have been observed in several countries [1]. Despite expanded screening programs, HCC is usually diagnosed at advanced stages (BCLC B or BCLC C, according to the Barcelona Clinic Liver Cancer (BCLC) Classification). These patients are ineligible for curative treatments like resection, ablation, or liver transplantation. Therefore, locoregional and systemic therapies are recommended treatment options in these cases. 

According to current guidelines, transarterial chemoembolization (TACE) is recommended as the standard of care for intermediate-stage HCC (BCLC B). In contrast, systemic therapy is reserved for advanced-stage HCC (BCLC C) [2]. In case of contraindications for TACE (e.g., portal vein thrombosis) or systemic treatment, the updated European Society for Medical Oncology (ESMO) Guidelines suggest Radioembolization (RE) as an alternative treatment option [3,4].

A retrospective study comparing the abovementioned locoregional therapeutic approaches showed longer time-to-progression (TTP) in RE-treated patients. The median overall survival was similar for both cohorts [5]. Furthermore, patients treated with RE suffered less often from abdominal pain and reported an overall better quality of life than those who received TACE [5,6]. Thus, RE represents an effective and well-tolerated alternative therapy for locally advanced HCC. However, careful patient selection is advisable in the case of borderline liver function since RE can result in hepatic deterioration even without evidence of radio-embolization-induced liver disease (REILD) or any prior signs of impaired liver function. In view of the increasing developments and new approvals, especially in the field of immune checkpoint-based systemic therapies, long-lasting liver function preservation is of great importance to enable a sequential therapy [7]. The investigation of additional tests or biomarkers to improve assessment of liver function and patient selection is therefore essential.

The liver maximum capacity (LiMAx^®^) test, a non-invasive breath test, was originally developed by Stockmann et al. as an additional tool to assess the risk of liver failure in hepatic surgery [8,9,10]. The test is based on the enzymatic metabolism of intravenously injected ^13^C-Methacetin by liver-specific cytochrome P450 (CYP) 1A2 and correlates well with overall functional and regenerative liver capacity [8,11].

While LiMAx^®^ has been well established in surgery, only a few studies have investigated its usefulness in the context of locoregional procedures. In a recently published prospective study, LiMAx^®^ was evaluated as an additional tool to predict the risk of hepatic failure related to TACE. The authors reported that lower LiMAx^®^ values at baseline (<150 µg/kg/h), representative of impaired liver synthesis function, were associated with a significant increase in serum bilirubin levels after TACE [12]. In another study, a considerable decrease in LiMAx^®^ could be observed one day after TACE, especially in patients with lower liver volume [13]. To the best of our knowledge, similar examinations have not yet been carried out in the context of RE.

In summary, RE is a valuable therapeutic option for locally advanced stages of HCC. However, patient selection is crucial for the success of the therapy. As the role of RE in this setting is unclear, we set out to investigate the predictive power of LiMAx^®^ in this work.

## 2. Patients and Methods

### 2.1. Ethics Approval Statement

The study was approved by the local ethics committee of the University Hospital Essen (reference number 16-7307-BO). The study has been performed following the 1964 Helsinki declaration. Each patient signed written informed consent for RE. For the prospective part of the study, the patients also signed informed consent for (repetitive) LiMAx^®^ measurement and the collection of personal data.

### 2.2. Study Population and Design

A total of 50 patients were assessed for the relationship between LiMAx^®^ and RE-related deterioration of hepatic function in advanced HCC. The study population was composed of a prospective and a retrospective cohort. The retrospective cohort included 13 patients with advanced HCC with or without liver cirrhosis who underwent RE at the University Hospital Magdeburg, Germany. The prospective part of the study included 37 patients who received RE at the University Hospital Essen, Germany.

Diagnosis of HCC was established following the criteria of the European Association for the Study of the Liver (EASL) [2]. We used multiphasic computed tomography (CT) or dynamic contrast-enhanced liver magnetic resonance imaging (MRI) as part of the staging examination before RE and 90 days after therapy to determine tumor response. A thoracic CT scan and skeletal scintigraphy were performed to detect possible extrahepatic tumor manifestations. The tumor stage was defined using the criteria of BCLC classification [14].

A multidisciplinary tumor board verified the indication and suitability for RE. According to current guidelines, patients were considered suitable for RE if they showed tumor progression following TACE or were unsuitable for systemic therapies. This includes both BCLC stage B patients and BCLC stage C patients with vascular infiltration or prognostically irrelevant metastases. Advanced extrahepatic tumor growth was an exclusion criterion for performing RE. Patients with an untreatable gastrointestinal or hepatopulmonary shunting and a tumor volume that occupies more than 70% of the liver were excluded from RE. In addition, total bilirubin of less than 2 mg/dL was required for performing RE. The study included neither patients who received RE with curative intent nor those who were bridged to liver transplantation. 

In addition to the Child–Pugh score (CPS), the albumin-bilirubin (ALBI) grade and model of end-stage liver disease (MELD) score were calculated to determine liver function. The liver function was determined before RE and 28 days and 90 days after treatment. 

In addition to laboratory parameters, the following clinical parameters were recorded: age, sex, body measurements (weight, height, and BMI), performance status according to Eastern Cooperative Oncology Group (ECOG-PS) criteria, and etiology of liver disease.

### 2.3. Liver Maximum Capacity (LiMAx^®^) Measurement

The substrate methacetin is exclusively metabolized by CYP1A2, a liver-specific enzyme. After intravenous injection of a body-weight-adjusted dose, ^13^C-labeled methacetin is oxidated into ^13^CO_2_ and paracetamol within the hepatocytes. For a period of 60 min after injection, breath samples are analyzed for the concentration of ^13^CO_2_ compared to the individual ^13^CO_2_/^12^CO_2_ baseline ratio, which was measured before methacetin injection. The ^13^CO_2_ concentration highly correlates with the metabolic capacity of the liver. The method was first described by Stockmann et al. to assess the risk of liver failure after hepatectomy [8]. LiMAx^®^ was performed using the LiMAx FLIP^®^ 2.0 detection device (Humedics, Berlin, Germany), following the manufacturer’s standard protocol. Results are provided as maximum metabolization capacity for ^13^C-Methacetin in µg/kg/h.

For retrospectively enrolled patients, LiMAx^®^ was only performed before the intervention (day 0). In the prospective part of the study, LiMAx^®^ was performed at day 0 as well as 28 days and 90 days after RE. 

### 2.4. Radioembolization

RE was performed using standard protocols. Briefly, after baseline angiography for precise therapy planning, the intervention was simulated using intraarterial ^99^mTc-labelled macro aggregated albumin (MAA) infusion. This sequence can identify relevant therapy-limiting shunts like hepatopulmonary or gastrointestinal shunts. Depending on tumor stage, treatment was performed as segmental, unilobar, or bilobar. Bilobar RE was performed in either one or two sessions, depending on liver function and tumor volume. For sequential bilobar RE, the second session was performed four weeks after the first intervention. CT-Scan or MRI was performed 12 weeks after completion of therapy to assess tumor response according to the modified Response Evaluation Criteria in Solid Tumors (mRECIST) criteria.

### 2.5. Statistical Analysis

Statistical analysis was performed using SPSS Statistics 27 (IBM SPSS Statistics version 27.0, Armonk, NY, USA) and Prism 9 (GraphPad Prism version 9.0, San Diego, CA, USA). The baseline characteristics were summarized using descriptive statistics. The data are presented as median with interquartile range (IQR). Continuous variables with normal distribution were compared using the student’s *t*-test (*t*-test) or the analysis of variance (ANOVA) or mixed-effect model for more than two variables. Mann–Whitney U test and Kruskal–Wallis test were used for non-normally distributed data. For the correlation analysis, we used Spearman’s rank correlation coefficient or Pearson correlation coefficient depending on whether the data were normally distributed or not. The diagnostic accuracy of different models to predict postinterventional hepatic decompensation was tested using the Area Under the Receiver Operating Characteristic (AUROC) curve. The Youden index (YI), defined as the sum of sensitivity and specificity minus one, determined the ideal cutoff. *p* values were two-sided with a significance level of 0.05. 

## 3. Results

### 3.1. Baseline Characteristics

For the present study, a total of 50 patients were enrolled. Of these, 37 patients were assigned to the prospective cohort. The baseline characteristics of the whole and the prospective cohort are summarized in Table 1. Regarding the whole cohort, most patients were male, with a median age of 70 years. The majority of patients included presented a good general condition (ECOG PS 0 n = 30, 61%; ECOG PS 1 n = 17, 35%; ECOG PS 2 n = 2, 4%) even though 82% of patients suffered from liver cirrhosis. The most common underlying liver diseases were alcoholic steatohepatitis (ASH, n = 18, 36%) and nonalcoholic steatohepatitis (NASH, n = 13, 26%). Only 12% of the patients tested positive for hepatitis B infection. At baseline, most patients showed a compensated liver function (94% with Child–Pugh score A; 60% with ALBI grade 1). Ascites was present in only three patients. No patient enrolled in the current study had a history of hepatic encephalopathy. BCLC B was the most frequently encountered tumor stage (70% with BCLC B and 26% with BCLC C). The median tumor volume was 205.1 mL (IQR 142.9–586.5 mL). The median baseline LiMAx^®^ value was 305 µg/kg/h (IQR 224–378.3 µg/kg/h). Most patients underwent one RE procedure only. 

### 3.2. Assessment of LiMAx^®^ and Liver Function at Baseline and 28 and 90 Days after RE

While patients without liver cirrhosis had normal baseline LiMAx^®^ values, patients with liver cirrhosis had baseline LiMAx^®^ values below the normal range. However, a significant difference could not be detected (with liver cirrhosis median 279 µg/kg/h, IQR 220–370.5 µg/kg/h; without liver cirrhosis median 363 µg/kg/h, IQR 291–389.5 µg/kg/h; *p* = 0.1464). For the prospectively included patients, LiMAx^®^ was also performed on day 28 and day 90 after RE. Regarding the whole prospective cohort, mixed-effect analysis did not reveal any significant deterioration of LiMAx^®^ at the given time point. However, in a subgroup analysis of patients with liver cirrhosis, LiMAx^®^ values were significantly lower 90 days after RE compared to baseline (d0 mean 312 µg/kg/h, d90 mean 217 µg/kg/g, *p* = 0.02). It must be noted that on day 90, LiMAx^®^ results were only available for 21 patients (Five patients had died by day 90; 11 patients did not undergo LiMAx^®^ on day 90) (Figure 1). Of the five patients who died during the three-month follow-up, only two presented hepatic deterioration. Their baseline LiMAx^®^ values were 363 µg/kg/h and 225 µg/kg/h, respectively. The patient with the lower baseline LiMAx^®^ value showed more severe hepatic impairment with an increase in bilirubin to 4 mg/dl and the occurrence of hepatic encephalopathy. Appendix A provides a detailed description of the causes of death.

As an established marker of liver function, serum levels of bilirubin and albumin, MELD score, and ALBI grade were measured on day 0, day 28, and day 90 and are summarized in Table 2. Regarding the whole cohort, most patients presented with ALBI grade 1 at baseline (ALBI grade 1: n = 30, 60% vs. ALBI grade 2: n = 19, 38%). The number of patients with ALBI grade 2 increased at day 28 (ALBI grade 1: n = 18, 40% vs. ALBI grade 2: n = 26, 58%). Ninety days after RE, there was no further increase in patients with an ALBI Grade 2. However, five patients had died at day 90. Using a mixed-effect model we could demonstrate a significant deterioration of the ALBI score at both day 28 and day 90 compared to baseline (Appendix A). The deterioration corresponds to the transition from ALBI grade 1 (baseline) to ALBI grade 2 (day 28 and d90) (Table 2 and Appendix A). Concerning the MELD Score, we observed a deterioration of one point at each predefined time point after RE (day 28, day 90). Serum bilirubin levels tended to deteriorate after RE, but there was no significant difference at the assessed time points (baseline 0.7 mg/dl, day 28 0.9 mg/dl, day 90 1.0 mg/dl, *p* = 0.07). Serum albumin values remained stable over the defined follow-up period. Regarding the established liver function parameters, a similar distribution could be observed for the prospectively examined patient group. We found a significant negative correlation between baseline LiMAx^®^ and bilirubin, ALBI grade, and MELD score at baseline and 28 days after RE. Additionally, LiMAx^®^ and albumin showed a positive correlation (Figure 2 and Appendix A). 

### 3.3. The Predictive Value of LiMAx for Hepatic Decompensation 28 and 90 Days after RE

Optimal patient selection is crucial for RE success and identifying patients at risk for hepatic decompensation. ALBI grade and Child–Pugh score represent established scores in depicting hepatic functional status. Therefore, we examined baseline LiMAx^®^ values of patients who presented a Child–Pugh Score A or B 28 days after RE. Eight patients had a Child–Pugh score B on day 28 after RE. In 38 patients, the Child–Pugh score remained at stage A (Table 2). For patients with Child–Pugh score B 28 days after RE, we could retrace significantly lower median LiMAx^®^ values at baseline (Child–Pugh score B: 220 µg/kg/h, IQR 176–227 µg/kg/h vs. Child–Pugh score A: 308 µg/kg/h, IQR 248.5–378.8 µg/kg/h, *p* = 0.007) (Figure 3).

ROC analysis was performed to assess how strongly LiMAx^®^ can discriminate between Child–Pugh scores A and B on day 28. In addition to LiMAx^®^, we included ALBI grade and MELD score at d0 (Figure 2). LiMAx^®^ test reached a similarly powerful AUC compared to MELD score and ALBI grade (LiMAx^®^ AUC 0.818, ALBI grade AUC 0.854, MELD score AUC 0.804). Youden’s Index was used to define the ideal cutoff. The cutoff values, including the sensitivity and specificity, are shown in Table 3. The ideal cutoff value for LiMAx^®^ was 229 µg/kg/h. 

In a further step, we stratified patients according to the difference in Child–Pugh score points between the baseline and day 28 and day 90 after RE (Figure 4 and Figure 5). Patients with a deterioration of at least one point in CPS (Δ CPS) at day 28 had significantly lower baseline LiMAx^®^ values (Δ CPS d28-d0 = 0 median baseline LiMAx^®^: 321 µg/kg/h, IQR 261–398 µg/kg/h, Δ CPS d28-d0 ≥ 1 median baseline LiMAx^®^: 226.5 µg/kg/h, IQR 200.3–308.3 µg/kg/h, *p* = 0.004) (Figure 4A). Secondly, we could observe that the LiMAx^®^ values at day 28 were significantly lower in patients where the Child–Pugh score deteriorated by at least one point from day 0 to day 28 (Δ CPS d28-d0 = 0 median baseline: LiMAx^®^: 342 µg/kg/h, IQR 277–420.5 µg/kg/h, Δ CPS d28-d0 ≥ 1 median baseline LiMAx^®^: 229 µg/kg/h, IQR 199.5–243.5 µg/kg/h, *p* = 0.02) (Figure 4B). 

Finally, we assessed if baseline LiMAx^®^ values were lower in patients that showed a deterioration of liver function at d90 of at least two points in Child–Pugh score. Here, a significant difference in baseline LiMAx^®^ could not be observed. (Δ CPS d90-d0 = 0–1 median baseline LiMAx^®:^ 311 µg/kg/h, IQR 243.5–420.5 µg/kg/h, Δ CPS d90-d0 ≥ 2 median baseline LiMAx^®^ 216 µg/kg/h, IQR 200.8–267 µg/kg/h, *p* = 0.05) (Figure 5A). Of note, in these patients showing a deterioration of liver function at day 90 by at least two points, LiMAx^®^ measured at day 28 was already significantly lower than in patients with rather stable liver function from day 0 to day 90 (Figure 5B).

## 4. Discussion

Immunotherapy, especially the combination of atezolizumab and bevacizumab, is considered the new gold standard for systemic therapy in HCC [15,16]. However, not all subgroups of HCC patients benefit from immunotherapy equally. For example, recent studies demonstrated that patients with non-viral HCC could benefit less from immunotherapy [7,17]. Predictors for optimizing patient selection are currently being studied with great effort, but cannot yet be used in everyday clinical practice [18]. Thus, there is a clear clinical need to improve the application and safety profile of non-systemic therapies (e.g., RE) as an alternative treatment for these patients. Three large randomized multicenter trials—SARAH, SIRveNIB, and SORAMIC—compared the outcome of RE to sorafenib (the gold standard in systemic therapy when our study was conducted) and could not demonstrate a significant survival benefit of RE over systemic therapy [19,20,21]. However, a meta-analysis including the results of these multicenter trials could demonstrate a non-inferior overall survival of RE compared to sorafenib. For patients without liver cirrhosis, even superiority of RE could be shown [22]. Of note, in the presence of NASH or HBV, HCC can occur in the absence of liver cirrhosis. Therefore, these patients could benefit from RE. Additionally, in a recently published retrospective analysis investigating the safety profile and survival benefit of RE for patients with HCC due to HBV infection compared to those presenting a NASH, an equal outcome was achieved for non-viral HCC patients [23]. Noteworthy, therapy-tolerability was significantly better for RE as compared to sorafenib, both in the SARAH and SIRveNIB trial [19,20].

A prerequisite for a favorable outcome of locoregional therapy in patients with advanced HCC is a preserved liver function [24]. HCC is frequently complicated by liver cirrhosis, entailing impaired liver function. However, even if liver cirrhosis is absent, liver function can be fragile, especially in patients suffering from NASH. Additionally, extensive tumor burden negatively impacts liver function due to decreased functional liver volume. Extensive tumor burden is often encountered in patients eligible for RE, thus putting these patients at high risk for further decrease in liver function.

To improve patient risk stratification before RE, we examined whether additional in-vivo liver function testing by LiMAx^®^ potentially facilitates clinical decision-making in these highly complex patients. A prospective analysis showed that LiMAx^®^ correlates well with the degree of fibrosis in patients with chronic liver disease [25]. LiMAx^®^ is already well established in the prediction of the functional reserve capacity of the liver before surgical resection [8,9,26]. In transplantation-medicine, LiMAx^®^ showed robust results in predicting the risk of graft dysfunction after liver transplantation [27]. Recently, LiMAx^®^ was also investigated as an additional tool for patient selection prior to transjugular intrahepatic portosystemic shunt implantation [28]. In HCC patients, LiMAx^®^ has shown promising results in assessing liver function after TACE and was more accurate than the Child–Pugh score [12,13]. 

Current therapy recommendations propose the absence of ascites and total bilirubin levels <2 mg/dL as a prerequisite for RE [24]. Despite complying with these criteria and presenting a Child–Pugh score A, some patients are still subject to hepatic decompensation after RE. In addition to the Child–Pugh score, the ALBI score is also a well-established test for assessing liver function. Unlike the Child–Pugh score, ALBI does not include any subjective parameters. This score notably facilitates an additional risk stratification within the Child–Pugh score A population [29]. In a retrospective analysis, patients with Child–Pugh score A and ALBI grade 2 were at substantially higher mortality risk due to hepatic dysfunction after RE than patients with ALBI grade 1 [30]. Previous studies confirmed ALBI score as a helpful predictor of hepatic deterioration after RE [30,31]. However, it still harbors several pitfalls in tumor patients, such as inflammation and malnutrition, that can undoubtedly influence relevant parameters such as albumin as an integral part of those scores. 

The majority of the patients in our cohort exhibited a well-preserved liver function before RE, corresponding to Child–Pugh score A and ALBI grade 1 with a total bilirubin level <2 mg/dL. Four weeks after therapy, impaired liver function, represented by Child–Pugh score ≥B7, was documented for some patients. In those patients, LiMAx^®^ values and ALBI score were worse at pre-therapeutical measurement. A one-point increase in Child–Pugh score on day 28 after RE was associated with worse baseline LiMAx^®^ values. Similar findings were derived from a prospective study, where LiMAx^®^ was evaluated as an additional tool to predict the risk of hepatic failure caused by TACE. The authors reported that lower LiMAx^®^ values at baseline (<150 µg/kg/h) were associated with a significant increase in serum bilirubin levels after TACE [12]. This shows that the LiMAx^®^ measurement could predict even a minor deterioration in liver function due to locoregional therapy. 

Interestingly, in our study, a deterioration in Child–Pugh score and ALBI score at day 28 was not paralleled by a decline in LiMAx^®^ at that point. However, three patients died during follow-up. Therefore, for these patients, no LiMAx^®^ results could be obtained on day 28, which leaves a possible bias. In a subgroup analysis only comprising patients with liver cirrhosis, we could find a significant decline in LiMAx^®^ at day 90, which could be due to an impaired ability of compensatory hypertrophy after RE in these patients. Thus, the functional reserve capacity is reduced in patients with liver cirrhosis [32,33]. While we only determined LiMAx^®^ values at baseline and four weeks after RE, Barzakova et al. investigated short-term alterations of LiMAx^®^. A significant decrease in LiMAx^®^ could be detected one day after TACE, especially in patients with lower liver volume before TACE. In contrast, baseline LiMAx^®^ was independent of the initial liver-volume [13]. The distinct deterioration in LiMAx^®^ after TACE was not associated with changes in the Child–Pugh score in this cohort, also supporting the assumptions that LiMAx^®^ might be more sensitive to subtle changes in liver function than the Child–Pugh score. 

Overlooking our results in the light of current literature, LiMAx^®^ seems to be able to predict already small changes in liver function after locoregional therapy and appears applicable as an additional tool for risk-stratification in patients with HCC before RE. Considering the increasing number of approved or investigated options of systemic therapy, the precise assessment and preservation of liver function, as well as the determination of the optimal timepoint for transition from locoregional to systemic therapy are essential [7,16,34,35]. LiMAx^®^ could add valuable insights to in vivo liver function for comprehensive clinical decision-making in a selected cohort of patients with borderline acceptable liver function before RE. Another interesting field of application might be the investigation of LiMAx^®^ as an additional tool for monitoring patients with Child–Pugh B liver function in the context of systemic therapy. 

Our study certainly holds several limitations that ought to be considered. Firstly, we only examined a limited number of patients. In total, 50 patients that were treated exclusively at German tertiary cancer centers were examined. Secondly, our participants represent a rather inhomogeneous sample, as different tumor stages might have caused variability in RE-protocols, certainly with different impact on liver function. An expansion of the study to a larger patient cohort could have reduced possible confounding factors regarding patient selection and would have enabled subgroup analyses with regard to different tumor stages and RE protocols.

## 5. Conclusions

Patients with low LiMAx values despite Child–Pugh A situation at baseline have a substantially increased risk for hepatic decompensation after RE. Therefore, LiMAx can be used as an additional tool to identify patients at high risk of post-interventional hepatic failure.

## Figures and Tables

**Figure 1 cancers-14-04584-f001:**
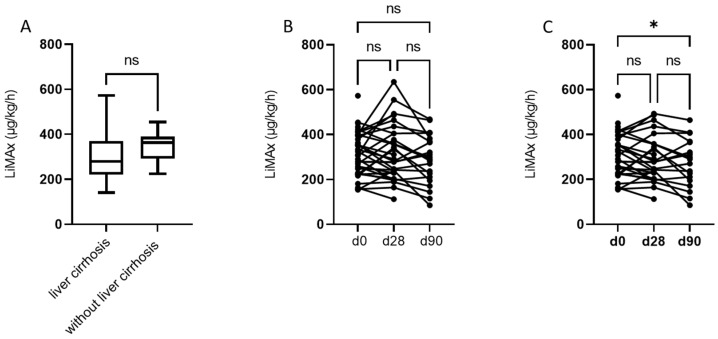
(**A**) Baseline LiMAx^®^ is not different in patients with or without liver cirrhosis (with liver cirrhosis median 279 µg/kg/h, IQR: 220–370.5 µg/kg/h; without liver cirrhosis median 363 µg/kg/h, IQR 291–389.5 µg/kg/h; *p* = 0.1464); (**B**) LiMAx^®^ at d0, d28, and d90 for the entire prospective cohort; (**C**) LiMAx^®^ at d0, d28, and d90 only for patients from the prospective cohort with liver cirrhosis. * = *p* < 0.05, ns = not significant.

**Figure 2 cancers-14-04584-f002:**
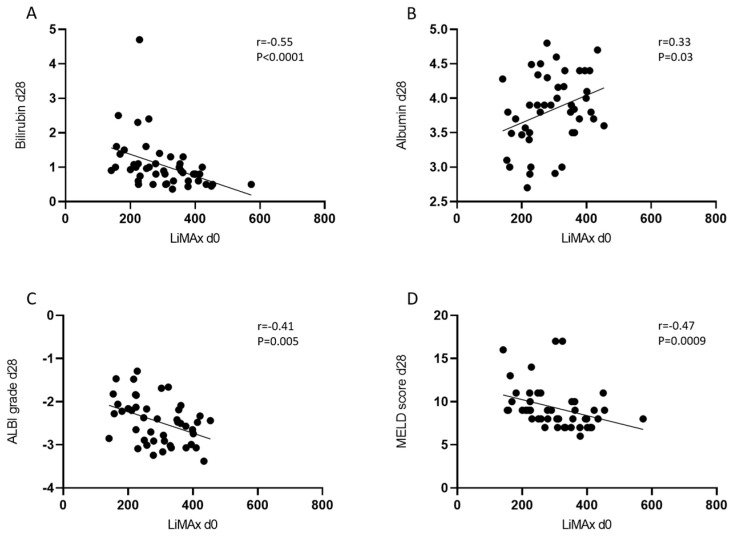
Correlation analysis of LiMAx^®^ at baseline with liver function 28 days after RE in the whole cohort. (**A**) bilirubin r = −0.55, *p* < 0.0001; (**B**) albumin r = 0.33, *p* = 0.03; (**C**) ALBI grade r = −0.41, *p* = 0.005; (**D**) MELD score r = −0.47, *p* = 0.0009.

**Figure 3 cancers-14-04584-f003:**
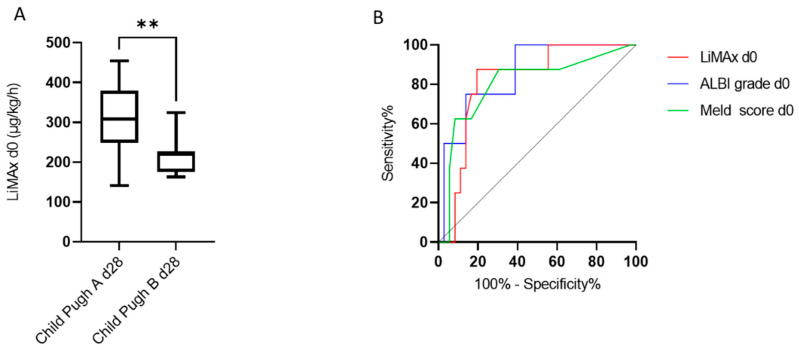
(**A**) Baseline LiMAx^®^ was significantly lower in patients presenting Child–Pugh B liver function 28 days after RE (Child–Pugh A median LiMAx 308 µg/kg/h, Child–Pugh B median LiMAx 220 µg/kg/h, *p* = 0.007); (**B**) Receiver operating characteristics (ROC) curve to predict the risk of hepatic decompensation in the whole cohort (AUC Meld score 0.803, AUC ALBI grade 0.854, AUC LiMAx 0.818), ** = *p* < 0.01.

**Figure 4 cancers-14-04584-f004:**
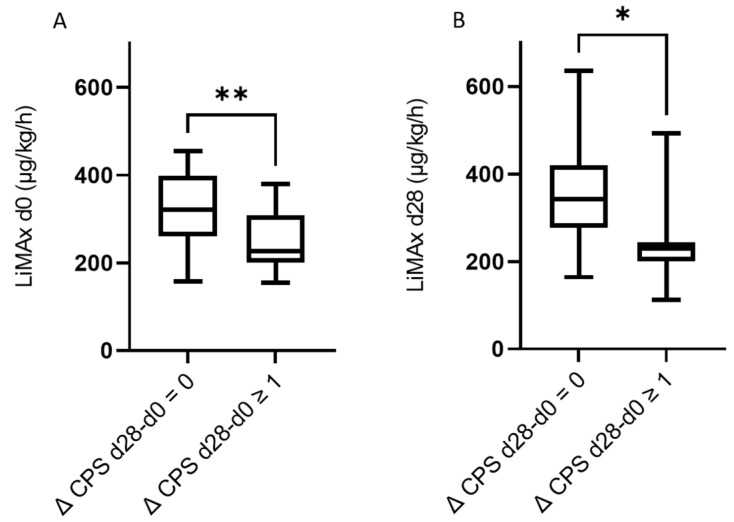
(**A**): Patients without any deterioration in Child–Pugh Score 28 days after RE presented significantly higher baseline LiMAx values, whole cohort (Δ CPS d28-d0 = 0 median baseline LiMAx 321 µg/kg/h, IQR 261–398 µg/kg/h, Δ CPS d28-d0 ≥ 1 median baseline LiMAx 226.5 µg/kg/h, IQR 200.3–308.3 µg/kg/h, *p* = 0.004); (**B**) Patients without any deterioration in Child–Pugh Score 28 days after RE presented significantly higher LiMAx values 28 days after RE, only prospective cohort (Δ CPS d28-d0 = 0 median baseline LiMAx 342 µg/kg/h, IQR 277–420.5 µg/kg/h, Δ CPS d28-d0 ≥ 1 median baseline LiMAx 229 µg/kg/h, IQR 199.5–243.5 µg/kg/h, *p* = 0.02). * = *p* < 0.05, ** = *p* < 0.01.

**Figure 5 cancers-14-04584-f005:**
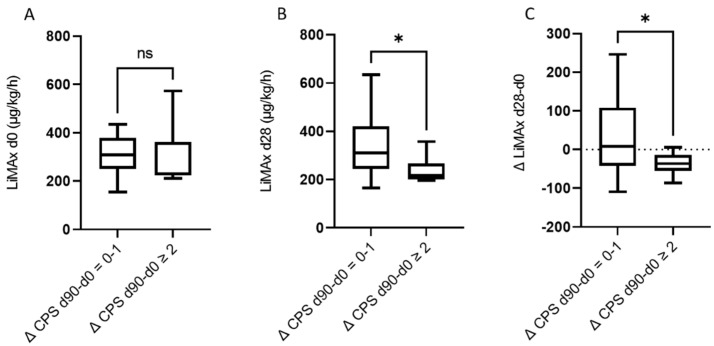
(**A**) No significant difference in baseline LiMAx for patients with or without deterioration in Child–Pugh score 90 days after RE. (**B**) Patients with at least two points decline in Child–Pugh score at d90 presented a significantly lower LiMAx at d28, only prospective cohort (Δ CPS d90-d0 = 0–1 median baseline LiMAx 311 µg/kg/h, IQR 243.5–420.5 µg/kg/h, Δ CPS d90-d0 ≥ 2 median baseline LiMAx 216 µg/kg/h, IQR 200.8–267 µg/kg/h, *p* = 0.05; (**C**) Patients without any deterioration in Child–Pugh score showed unchanged LiMAx values at d28. Patients with a deterioration in Child–Pugh score by at least two points at day 90 had slightly worsened LiMAx values, only prospective cohort. * = *p* < 0.05, ns = not significant.

**Table 1 cancers-14-04584-t001:** Baseline characteristics.

	Parameter/Grade/Stage	Whole Cohort (N = 50)	Prospective Cohort (N = 37)
Age at first SIRT	years	70 (64–75)	68 (64–73)
Sex	Male/female	44 (88)/6 (12)	33 (89)/4 (11)
ECOG PS ^a^	0	30 (61)	26 (70)
	I	17 (35)	10 (27)
	II	2 (4)	1 (3)
Cirrhosis		41 (82)	32 (87)
Etiology	ASH	18 (36)	14 (38)
	NASH	13 (26)	7 (19)
	Hepatitis B	6 (12)	5 (14)
	Hepatitis C	3 (6)	3 (8)
	Hemochromatosis	2 (4)	2 (5)
	Other	8 (16)	6 (16)
Child–Pugh score	A	47 (94)	35 (95)
	B	3 (6)	2 (5)
ALBI grade	1	30 (60)	22 (60)
	2	19 (38)	15 (40)
	3	1 (2)	0
Ascites	Absent Grade I-II	47 (94) 3 (6)	34 (92) 3 (8)
Encephalopathy	history	0	0
BCLC	A	2 (4)	0
	B	35 (70)	26 (70)
	C	13 (26)	11 (30)
Macrovascular invasion		13 (26)	7 (19)
Portal vein thrombosis	Main Branch	3 (6) 2 (4)	3 (8) 1 (3)
Tumor manifestations	Unilobar	17 (34)	10 (27)
	Bilobar	33 (66)	27 (73)
Number of lesions ^a^	1	9 (18)	7 (19)
	1–3	4 (8)	2 (5)
	>3	36 (73)	28 (76)
Tumor volume, mL		205.1 (142.9–586.5)	174.5 (127.4–339.0)
Number of RE sessions	1 2	33 (66) 17 (34)	24 (65) 13 (35)
LiMAx^®^, µg/kg/h	d0 d28 d90	305 (224–378.3)	309 (226.5–398) 286 (222.5–405) 290 (203–375.5)

Data are presented as N (%), median (interquartile range), ECOG-PS, Eastern Cooperative Oncology Group performance status; ALBI grade, Albumin-Bilirubin grade; BCLC, Barcelona Clinic Liver Cancer Classification; RE, Radioembolization; LiMAx^®^, Liver Maximum capacity; d, day; ^a^ only 49 patients included; deviations from 100% are due to rounding.

**Table 2 cancers-14-04584-t002:** Liver function at baseline, day 28 and day 90.

Parameter		Whole Cohort			Prospective Cohort		
		Baseline N = 50	day28 N = 44	day 90 N = 43	Baseline N = 37	day 28 N = 33	day 90 N = 31
Child–Pugh score	A	47 (94)	36 (82)	32 (74)	35 (95)	27(82)	23(74)
	B	3 (6)	8 (18)	5 (12)	2 (5)	6 (18)	3 (10)
	C	0	0	1 (2)	0	0	1 (3)
	deceased	n.a.	0	5 (12)	n.a.	0	4 (13)
ALBI grade	1	30 (60)	18 (40) ^a^	17 (40)	22 (60)	13 (38) ^b^	12 (39)
	2	19 (38)	26 (58) ^a^	20 (47)	15 (40)	20 (59) ^b^	14 (45)
	3	1 (2)	1 (2) ^a^	1 (2)	0	1 (3) ^b^	1 (3)
	deceased	n.a.	n.a.	5 (12)	n.a.	0	4 (13)
Meld score		8 (7–10)	9 (8–10)	10 (8–12)	8 (7–9)	9 (8–10)	10 (8–12)
Bilirubin	mg/dL	0.7 (0.5–1.1)	0.9 (0.6–1.1)	1.0 (0.6–1.4)	0.8 (0.5–1.2)	0.95 (0.6–1.3)	1.2 (0.7–1.6)
Albumin	g/dL	4.1 (3.7–4.4)	3.8 (3.5–4.3)	4.0 (3.6–4.3)	4.1 (3.8–4.4)	3.9 (3.5–4.3)	4.0 (3.5–4.3)

Data are presented as N (%), median (interquartile range); ^a^ N = 45; ^b^ N = 34; ALBI grade, Albumin-Bilirubin grade; MELD, Model of End Stage Liver Disease; deviations from 100% are due to rounding.

**Table 3 cancers-14-04584-t003:** Predictive value of LiMAx, ALBI grade, and MELD score for hepatic decompensation (defined as CPS B7 or more) 28 days after RE (whole cohort).

Parameter	Youden Index	Sensitivity (%)	Specificity (%)	AUC
LiMAx^®^ day 0	229	87.50	80.56	0.818
ALBI grade day 0	−2.430	75.00	86.11	0.854
MELD score day 0	8.5	87.50	69.44	0.804

## Data Availability

Data will be made available by the corresponding author upon reasonable request.

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
