# Peer review of "LiMAx Prior to Radioembolization for Hepatocellular Carcinoma as an Additional Tool for Patient Selection in Patients with Liver Cirrhosis"

_cancers, 2022, doi:10.3390/cancers14194584_

Round 1

Reviewer 1 Report

I appreciate the opportunity to review this manuscript. The authors tried to use the LiMAx to predict the risk of hepatic decompensation after radioembolization. The results showed the predictive value was as well as ALBI grade and MELD score at baseline. The clinical application of this study is good and can apply an useful tool for clinicians before radioembolization. I recommend revising below-mentioned point is addressed. 

Q1: Can the authors describe the cause of death for the five patients died during the follow-up? Does the hepatic failure happen in these patients and what’s the baseline LiMax in these patients? 

Q2: The values of LiMax of some patients show increase in d28 and decrease in d90 in Fig 1B and 1C. How to explain this phenomenon? Does the ALBI score also show no difference at the d0, d28 and d90?

Author Response

Cancers - Manuscript ID: cancers-1914386

Title of Manuscript: LiMAx® prior to radioembolization for hepatocellular carcinoma as an additional tool for patient selection in patients with liver cirrhosis

Point-by-point reply to the reviewers

Reviewer #1:

Q1: “Can the authors describe the cause of death for the five patients died during the follow-up? Does the hepatic failure happen in these patients and what’s the baseline LiMAx in these patients?”

We agree that this is a piece of important information. Listed below are the causes of death and the corresponding baseline LiMAx® values of the five patients that died during follow-up. Briefly, of the five patients, only two (#25 and #40) presented a hepatic deterioration. Their baseline LiMAx® values were 363 µg/kg/h and 225 µg/kg/h, respectively. The patient with the lower baseline LiMAx® value showed more severe hepatic impairment with increased bilirubin and hepatic encephalopathy. We added this information as a supplementary table (see SI table 1)

Specific description regarding causes of death:

  • Patient #25: This patient showed a deterioration in liver function in terms of hydropic decompensation but without relevant bilirubin, albumin, and INR changes. Furthermore, significant tumor growth with extrahepatic tumor manifestations was recorded. The baseline LiMAx® value was 363 µg/kg/h.
  • Patient #36: This patient died due to pneumonia and new onset lung metastases. The baseline LiMAx® value was 573 µg/kg/h.
  • Patient #40: This patient showed a deterioration in liver function with an increase of total bilirubin to 4 mg/dl and the development of hepatic encephalopathy. However, the degree of hepatic encephalopathy cannot be determined from the digital CRF. The baseline LiMAx® was reduced and amounted to 225 µg/kg/h.
  • Patients #37 and #68: These two patients showed a deterioration in general condition. Their respective baseline LiMAx® values were 324 µg/kg/h and 225 µg/kg/h.

SI Table 1: Causes of death during follow-up.

Patient

Causes of death

Baseline LiMAx® value (µg/kg/h)

#25

Hydropic decompensation but without relevant changes of bilirubin, albumin, and INR. Relevant tumor growth with the occurrence of extrahepatic tumor manifestations

363

#36

Pneumonia and new onset lung metastases

573

#40

Hepatic deterioration with an increase of total bilirubin to 4 mg/dl and hepatic encephalopathy

225

#37

Deterioration in general condition

324

#68

Deterioration in general condition

225

Q2.1: “The values of LiMAx of patients show increase in d28 and decrease in d90 in Fig 1B and 1C. How to explain this phenomenon?”

We thank the author for this precise observation, and unfortunately, we cannot provide a final answer to these dynamics of LiMAx® in this subset of patients. However, one could speculate that the explanation for this phenomenon lies within the limitations of LiMAx® itself. One possible explanation might be the reliance of LiMAx® on patient cooperation (e.g., omission of smoking and strict adherence to the applied fasting protocol). Another explanation could be that radioembolization-induced hypertrophy or (overshoot)-regeneration after acute liver-damage results in a short-term improvement in metabolic capacity, which may be reflected by an increase in LiMAx®. A similar observation was made by Barzakova and colleagues, who investigated LiMAx® in the context of transarterial chemoembolization. While the LiMAx® values dropped by 10% on the first postinterventional day, they had again reached the baseline in the 1-month follow-up. The decline of the LiMAx values 90 days after RE could be explained by the general deterioration of liver function due to the progressive underlying liver disease but maybe also by a possible deterioration due to Radioembolization-induced liver disease (REILD).

Q2.2: “Does the ALBI score show no difference at the d0, d28, and d90?“

Indeed, compared with LiMAx®, the ALBI score showed a significant deterioration at 28 and 90 days after RE compared to baseline. However, from day 28 to day 90, no further significant decline was observed. We added this observation in the resubmitted manuscript.

SI figure 1: ALBI score at d0, d28, and d90 for the entire prospective cohort. ALBI score at d0, d28, and d90 only for patients with liver cirrhosis from the prospective cohort.

Reviewer 2 Report

The study assesses a current, timely topic.
We recommend some changes:
- We believe this article is suitable for publication in the journal although major revisions are needed. The main strengths of this paper are that it addresses an interesting and very timely question and provides a clear answer, with some limitations. Certainly, the study is limited to an Asian population with a very small sample size, and authors should further express this point.
- Second, the study included a widely varied patient population from a german institute and the total number of patients analyzed was relatively small. Thus, the authors should better highlight the limitations of the current paper.
- The background of the changing scenario of medical treatment in HCC should be better discussed, and some recent papers regarding this topic should be included (PMID: 34431725 ; PMID: 29968763 ; 
PMID: 35403533 ).
Major changes are necessary.

Author Response

Cancers - Manuscript ID: cancers-1914386

Title of Manuscript: LiMAx® prior to radioembolization for hepatocellular carcinoma as an additional tool for patient selection in patients with liver cirrhosis

Point-by-point reply to the reviewers

Reviewer #2:

“We believe this article is suitable for publication in the journal although major revisions are needed. The main strengths of this paper are that it addresses an interesting and very timely question and provides a clear answer, with some limitations.”

We thank the reviewer for this comprehensive analysis of our work.

Q1: “Certainly, the study is limited to an Asian population with a very small size, and the authors should further express this point. Secondly, the study included a widely varied patient population from a german institute and the total number of patients analyzed was relatively small. Thus, the authors should better highlight the limitations of the current paper.”

We fully agree with the reviewer that this is an essential limitation of our study. We acknowledge that 50 patients in our study represent a relatively small sample. In addition, as the reviewer points out, it is important to mention that the patients were exclusively treated at two German tertiary cancer centers (University hospitals of Magdeburg and Essen). Furthermore, there is evident heterogeneity in the study cohort concerning the tumor stage, leading to variable RE protocols with different impacts on liver function. Nevertheless, our study provides a basis for future prospective studies investigating the possibilities of LiMAx® for patient selection in the field of locoregional therapy. We took the opportunity to outline these points more clearly in the discussion.

Q2: “The background of the changing scenario of medical treatment in HCC should be better discussed and some recent paper regarding this topic should be included.“

We thank the reviewer for providing us with very interesting literature that enabled us to give the readers of our manuscript a broader view of the changing field of HCC therapy.

Round 2

Reviewer 2 Report

acceptance.